# HYPERGAN: GENERATING DIVERSE, PERFORMANT NEURAL NETWORKS WITH A GAN

## ABSTRACT

We introduce HyperGAN, a generative network that learns to generate all the weight parameters of deep neural networks. HyperGAN first transforms low dimensional noise into a latent space, which can be sampled from to obtain diverse, performant sets of parameters for a target architecture. We utilize an architecture that bears resemblance to generative adversarial networks, but we evaluate the likelihood of samples with a classification loss. This is equivalent to minimizing the KL-divergence between the generated network parameter distribution and an unknown true parameter distribution. We apply HyperGAN to classification, showing that HyperGAN can learn to generate parameters which solve the MNIST and CIFAR-10 datasets with competitive performance to fully supervised learning, while learning a rich distribution of effective parameters. We also show that HyperGAN can also provide better uncertainty than standard ensembles. This is evaluated by the ability of HyperGAN-generated ensembles to detect out of distribution data as well as adversarial examples. We see that in addition to being highly accurate on inlier data, HyperGAN can provide reasonable uncertainty estimates.

## 1 INTRODUCTION

Since the inception of deep neural networks, it has been found that it is possible to train from different random initializations and obtain networks that, albeit having quite different parameters, achieve quite similar accuracy (Freeman & Bruna, 2016). It has further been found that ensembles of deep networks that are trained in such a way have significant performance advantages over single models (Maclin & Opitz, 2011), similar to the classical bagging approach in statistics. Ensemble models also have other benefits, such as being robust to outliers and being able to provide variance or uncertainty estimates over their inputs (Lakshminarayanan et al., 2017).

In Bayesian deep learning, there is a significant interest in having a probabilistic interpretation of network parameters and modeling a distribution over them. Earlier approaches mostly utilize dropout as a Bayesian approximation, by randomly setting different parameters to zero and thus integrating over many possible networks. (Gal & Ghahramani, 2016) showed that networks with dropout following each layer are equivalent to a deep Gaussian process (Damianou & Lawrence, 2013) marginalized over its covariance functions. They proposed MCdropout as a simple way to estimate model uncertainty. These approximations are not well aligned with current training patterns of neural networks. Applying dropout to every layer results in over-regularization and underfitting of the target function. Moreover, dropout does not integrate over the full variation of possible models, only those which may be reached from one (random) initialization.

As another interesting direction, hypernetworks (Ha et al., 2016) are neural networks which output parameters for a target neural network. The hypernetwork and the target network together form a single model which is trained jointly. The original hypernetwork produced the target weights as a deterministic function of its own weights, but Bayesian Hypernetworks (BHNs) (Krueger et al., 2017), and Multiplicative Normalizing Flows (MNF) (Louizos & Welling, 2016) generate model parameters by sampling a Gaussian prior. However, these approaches use normalizing flows to transform a simple prior into a sample of the more complicated posterior, which are composed of only bijective, invertible functions. This limits their scalability and the variety of learnable functions.

In this paper we explore an approach which focuses on generating all the parameters of a neural network, without assuming any fixed noise models on parameters. To keep our method scalable, we avoid utilizing invertible functions as in normalizing flow approaches, and instead utilize the ideas from generative adversarial networks (GANs). We especially observe recent Wasserstein Auto-encoder (Tolstikhin et al., 2017) approaches. These approaches have demonstrated an impressive capability to model complicated, multimodal distributions. In our approach, a random noise vector is first encoded to a number of different random vectors, then each random vector is used to generate all parameters within one layer of a deep network. The generator is then trained with conventional maximum likelihood (classification/regression) on the parameters it generates, and an adversarial regularization keeps it from collapsing onto only one mode. In this way, it is possible to generate much larger networks than the dimensionality of the latent code, making our approach capable of generating all the parameters of a deep network with a single GPU. As an example, in our experiments on CIFAR-10 we start from a 256-dimensional latent vector and generate all $50,000+$ parameters in one pass, consuming only 4GB GPU memory. This shows that deep networks may indeed span a low-dimensional manifold, and could spur further thoughts and research.

## 1.1 SUMMARY OF CONTRIBUTIONS

We propose HyperGAN, a novel approach for generating all the parameters for a target network architecture using a modified GAN, and we do so starting from a small Gaussian noise vector which scales well with the size of the output. Our approach is different from strict Bayesian approaches since we do not attempt to model the entire posterior, i.e., from an observed network we cannot compute its probability of it being generated. Moreover, we use deterministic generators, our only source of stochasticity is the initial prior sample. After our GAN is trained, one can directly generate many diverse, well-trained deep models without needing to further train or fine-tune them. The diversity of the models we can generate is beyond just adding dropout or scaling factors, which is shown by the superior performance of ensembles of the generated networks.

We believe HyperGAN is widely applicable to a variety of tasks. One area where populations of diverse networks show promise is in uncertainty estimation and anomaly detection. We show through a variety of experiments that populations of networks sampled from HyperGAN are able to approximate the data distribution such that it can detect out of distribution samples. We show that we can provide a reasonable measure of uncertainty by calculating the entropy within the predictive distribution of sampled networks. Our method is straightforward, as well as easy to train and sample from. We hope that we can inspire future work in estimation of the manifold of neural networks.

## 2 RELATED WORK

Generating parameters for neural networks has been framed in contexts other than the Bayesian approaches described above. The hypernetwork framework (Ha et al., 2016) has come to describe models where one network directly supervises the weight updates of another network. (Pawlowski et al., 2017) uses hypernetworks to generate layer-wise weights for a target architecture given some auxiliary noise as input. The auxiliary noise is noted to be independent between layers, while we impose structure in our inputs to create diversity in our generated samples.

Computer vision methods often use data driven approaches as seen in methods such as Spatial Transformer networks (Jaderberg et al., 2015), or Dynamic Filter networks (Jia et al., 2016). In these methods the filter parameters of the main network are conditioned on the input data, receiving contextual scale and shift updates from an auxiliary network. Our method, however, generates parameters for the whole network. Furthermore our predicted parameters are highly nonlinear functions of the input, instead of simple affine transformations based on the input examples.

Recently, (Lakshminarayanan et al., 2017) proposed Deep Ensembles, where adversarial training was applied to standard ensembles to smooth the predictive variance. However, adversarial training is an expensive training process. Adversarial examples are generated for each batch of data seen. We seek a method to learn a distribution over parameters which does not require adversarial training.

Meta learning approaches use different kinds of weights in order to increase the generalization ability of neural networks. The first proposed method is fast weights (Hinton & Plaut, 1987) which uses an auxiliary (slow) network to produce weight changes in the target (fast) network, acting as a

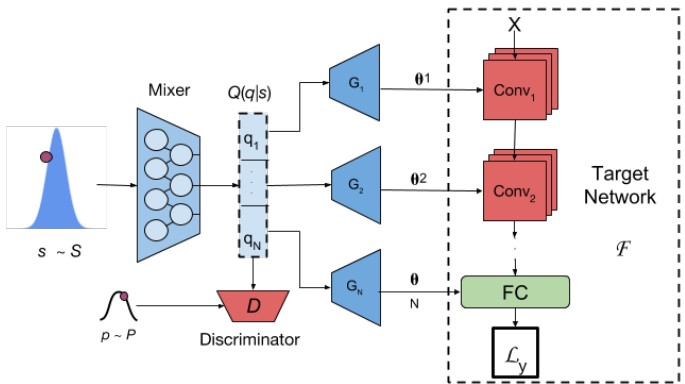

Figure 1: HyperGAN architecture. The mixer transforms $s \sim \mathcal{S}$ into latent codes $\{q_1, \ldots, q_N\}$. The generators each transform a latent subvector $q_i$ into the parameters of the corresponding layer in the target network. The discriminator forces $Q(q|s)$ to be well-distributed and close to $\mathcal{P}$

short term memory store. Meta Networks (Munkhdalai & Yu, 2017) build on this approach by using an external neural memory store in addition to multiple sets of fast and slow weights. (Ba et al., 2016) augment recurrent networks with fast weights as a more biologically plausible memory system than Neural Turing Machines. Unfortunately, the generation of each predicting network requires querying the base (slow) learner many times. Many of the methods presented there, along with hyperparameter learning (Lorraine & Duvenaud, 2018), and the original hypernetwork, propose learning target weights which are deterministic functions of the training data. Our method instead captures a distribution over parameters, and provides a cheap way to directly sample full networks from a learned distribution.

## 3 HYPERGAN

Taking note from the original hypernetwork framework for generating neural networks (Ha et al., 2016), we coin our approach HyperGAN. Standard GAN training dictates that we train a generator $G$ to model the target distribution, aided by a discriminator network $D$. This entails acquiring a large dataset of trained neural network parameters, and providing samples to $D$ to judge the output of the generator. However, a large collection of neural networks would be extremely costly to build. Instead, our data distribution is built online by evaluating the performance of our generated parameters at each step.

We begin with assuming that a neural network $\mathcal{F}(x; \theta)$ with input $x$ and parameters $\theta$ consisting of a given architecture with $N$ layers, and a training set $M$ with inputs and targets $(X, Y) = \{x_i, y_i\}_{i=1}^{M}$. The standard training regime consists of computing a loss function $\mathcal{L}(\mathcal{F}(x; \theta), y)$ and updating the parameters $\theta$ with backpropagation until $\mathcal{L}$ is minimized. This works if we only want a point estimate of $\theta$. However, if we want to generate more than one non-trivial network, some diversity is needed. Our approach to creating diversity is similar to a regular GAN architecture in that we start by drawing a random sample $s \sim \mathcal{S} = \mathcal{N}(\mathbf{0}, \mathbf{I}_j)$, where 0 is an all-zero vector and $\mathbf{I}_j$ is a $j \times j$ identity matrix, and generate parameters using a generator from $s$.

Figure 1 shows the HyperGAN architecture. Distinct from the standard GAN, we use parallel, untied generators to form the parameters of each layer. In order for that to work, we observe that parameters between network layers are not independent, but are strongly correlated. The parameters of each layer depend on the output and the parameters of the preceding layers. Therefore, the generated parameters must be correlated to produce well-performing neural networks. We propose to add a *Mixer* $Q$ which maps $s \sim \mathcal{S}$ to the mixed latent space $Z \in \mathbb{R}^{Nd}$, $d < j$. An $Nd$-dimensional $Q(s)$ is then partitioned into $N$ layer embeddings $[q_1, \ldots, q_N]$, each being a $d$-dimensional vector. Finally, we use $N$ parallel generators $G = \{G_1(q_1) \ldots G_N(q_n)\}$ to generate the parameters $\theta_G$ for all layers in $\mathcal{F}$.

After generation, we can evaluate the new model $\mathcal{F}(x, \theta_G)$ on the training set. We define an objective which minimizes the error of generated parameters with respect to a task loss $\mathcal{L}$:

$$\inf_{G,Q} \mathbb{E}_{s \sim \mathcal{S}} \mathbb{E}_{(x,y) \sim (X,Y)} \left[ \mathcal{L}(\mathcal{F}(x; G(Q(s))), y) \right] \tag{1}$$

At each training step we generate a different network $G(Q(s))$ from a random $s \sim S$, and then evaluate the loss function on a mini-batch from the training set. The resulting loss is backpropagated through the generators until $\theta_G$ minimizes the target loss $\mathcal{L}$.

The formulation in (1) presents a problem: the codes sampled from $Q(s)$ will certainly collapse to the maximum likelihood (ML) estimate (when $\mathcal{L}$ is a log-likelihood). This means that the generators will learn a very narrow approximation of $\Theta$, and indeed we see this happen in tables 3 and 4. To assure that the parameters are well distributed, we added an adversarial constraint on the mixed latent space $\mathcal{D}(Q(s))$ so that it should not deviate too much from a Gaussian prior $\mathcal{P}$. This constraint makes it closer to the generated parameters and ensures that $Q(s)$ itself does not collapse to always outputting the same latent code. With this we arrive at the HyperGAN objective:

$$\inf_{G,Q} \mathbb{E}_{s \sim \mathcal{S}} \mathbb{E}_{(x,y) \sim (X,Y)} \left[ \mathcal{L}(\mathcal{F}(x; G(Q(s))), y) \right] - \beta \mathcal{D}(Q(s), \mathcal{P}) \tag{2}$$

Where $\beta$ is a hyperparameter, and $\mathcal{D}$ is the regularization term which penalizes the distance between the prior and the distribution of latent codes. In practice $\mathcal{D}$ could be any distance function between two distributions. We choose to parameterize $\mathcal{D}$ as a discriminator network $D$ that output probabilities, and use the adversarial loss (Goodfellow et al., 2014) to approximate $\mathcal{D}(Q(s), \mathcal{P})$. Note that while $\mathcal{P}$ and $\mathcal{S}$ are both multivariate Gaussians, they are distinct distributions as seen in figure 1.

$$\mathcal{D} := -\sum_{i=1}^{N} \left( \log D(p_i) + \log(1 - D(q_i)) \right) \tag{3}$$

Note that we find it difficult to learn a discriminator in the output (parameter) space because the dimensionality is high and there is no structure in those parameters to be utilized as in images (where CNNs can be trained). Our experiments show that regularizing in the latent space works well. We hypothesize this is because of overparametrization in $\theta$. The latent space initializes with random projections, which have a restricted isometry property according to the Johnson-Lindenstrauss lemma, hence if the mixed latent factor $q$ is Gaussian, a random projection from it preserves the distance and diversity from $q$ to $\theta$. If $\theta$ is indeed severely overparametrized, and it is possible to generate diverse parameters by maximizing the likelihood, then the generators would not collapse to a single $\theta$, since that would require breaking the restricted isometry from the initialization.

This framework is general and can be adapted to a variety of tasks and losses. In this work, we show that HyperGAN can operate in both classification and regression settings. For multi-class classification, the generators and mixer are trained with the cross entropy loss function:

$$\mathcal{L}_H = M^{-1} \sum_{i=1}^{M} y_i \log \left( \mathcal{F}(x_i; \theta) \right) \quad \text{where} \quad \theta = \{ G_1(q_1), \ldots, G_n(q_n) \} \tag{4}$$

For regression tasks we replace the cross entropy term with the mean squared error function:

$$\mathcal{L}_{mse} = M^{-1} \sum_{i=1}^{M} (y_i - \mathcal{F}(x_i; \theta))^2 \quad \text{where} \quad \theta = \{ G_1(q_1), \ldots, G_n(q_n) \} \tag{5}$$

### 3.1 LEARNING WITHOUT AN EXPLICIT TARGET DISTRIBUTION

In implicit generative models such as GAN or WAE (Tolstikhin et al., 2017), it is always necessary to have a collection of inlier data points to train with. Inlier samples come from the distribution that is being estimated, and learning without them is difficult. HyperGAN does not have a given set of inlier points to train with. Instead, HyperGAN uses the error on the target loss to estimate the distance between the generated samples $\theta_G \in \Theta_G$ and the target samples $\theta \in \Theta$. We note that $\theta_G$ represents the maximum likelihood estimate of $\mathcal{F}(x; \theta)$. (6) shows that by minimizing the error of the ML estimate on the log-likelihood, we are in fact minimizing the KL divergence between the target distribution $\Theta$ and the generated samples $\theta_G \in \Theta_G$.

$$\begin{aligned} \inf_{\theta_G} D_{KL}(P(x|\theta) || P(x|\theta_G)) &= \inf_{\theta_G} \mathbb{E}_{P(x|\theta_G)} \left[ \log P(x|\theta) - \log P(x|\theta_G) \right] \\ &= \inf_{\theta_G} \mathbb{E}_{P(x|\theta_G)} \left[ -\log P(x|\theta_G) \right] \end{aligned} \tag{6}$$

Where the term $\log P(x|\theta)$ is the entropy of the target distribution and does not contribute to the optimization problem. The derived likelihood function is flexible and used widely across different domains. In our work it is represented by the cross entropy and MSE losses that we study.

Its useful to note here how HyperGAN is distinct from other density estimators such as GAN and WAE. These methods use a likelihood function, such as MSE, to estimate the quality of the samples (in GANs the likelihood function is learned via the discriminator). The likelihood is given as the distance from generated samples, to given samples of the target distribution. HyperGAN instead estimates both the target and the approximation. We assume the target distribution exists, and update our approximation $\Theta_G$ to better match $\Theta$. If we have samples of the target distribution, we can reduce HyperGAN to a GAN (by moving the discriminator to the output space).We thoroughly explore the connections and differences to GANs and WAE in A.5. HyperGAN is different from fully probabilistic approaches such as MNF (Louizos & Welling, 2016) since one cannot compute the probability of the generated $\theta$, nor can one encode a generated $\theta$ back to the latent spaces.

# 4 EXPERIMENTS

## 4.1 HIGH LEVEL DESCRIPTION AND EXPERIMENTAL SETUP

We conduct a variety of experiments to show HyperGAN's ability to achieve both high accuracy and obtain accurate uncertainty estimates. First we show classification performance on both MNIST and CIFAR-10 datasets. Next we examine HyperGAN's ability to learn the variance of a simple 1D dataset. We perform experiments on anomaly detection: testing HyperGAN on notMNIST, and 5 classes of CIFAR-10 which are hidden during training. We also examine adversarial examples as extreme cases of off-manifold data, and test our robustness to them. In our experiments we compare with (Louizos & Welling, 2016) (MNF) as well as standard ensembles.

In all experiments we report results with two HyperGANs, one trained on MNIST and another on CIFAR-10. Both of our models take a 256 dimensional sample of $\mathcal{S}$ as input, but have different sized mixed latent spaces. The HyperGAN for the MNIST experiments consists of three weight generators, each using a 128 dimensional latent point as input. The target network for the MNIST experiments is a small two layer convolutional network, using leaky ReLU activations and 2x2 max pooling after each convolutional layer. Our HyperGAN trained on CIFAR-10 used 5 weight generators and latent points with dimensonality 256. The target architecture for CIFAR-10 tests consists of three convolutional layers, each followed by leaky ReLU and 2x2 max pooling. The exact architectures we used for the target networks is given in A.4. In our experiments we used the same network architecture across different methods. Note that our architecture is different from the LeNet-5 used in MNF, yet we see similar results as reported in (Louizos & Welling, 2016).

### HYPERGAN DETAILS

The mixing network, generators, and discriminator are each parameterized by 2 layer MLPs with 512 units each and ReLU nonlinearities. We found in a pilot study that larger networks offered little performance benefit, and ultimately hurt scalability. In all experiments, we pretrain the mixer so that the mean and covariance of $Q(q|s)$ are close to $\mathcal{P}$. It should be noted that the architecture is flexible. The number and width of layers may be varied without harming HyperGANs ability to model the target distribution. We trained our HyperGAN on MNIST using less than 1.5GB of memory on a single GPU, while CIFAR-10 used just 4GB, making HyperGAN surprisingly scalable.

In Table 1 we show some statistics of the networks generated by HyperGAN on MNIST. We note that HyperGAN can generate very diverse networks, as the variance of network parameters generated by the HyperGAN is significantly higher than standard training from different random initializations.

## 4.2 CLASSIFICATION

First we evaluate the classification accuracy of HyperGAN on MNIST and CIFAR-10. Classification serves as an entrance exam into our other experiments, as the distribution we want to learn is over parameters which can effectively solve the classification task. We test with both single network samples, and ensembles. For our ensembles we average predictions from $N$ sampled models with

|  | HyperGAN | | | Standard Training | | |
|---|---|---|---|---|---|---|
|  | Conv1 | Conv2 | Linear | Conv1 | Conv2 | Linear |
| Mean | 7.49 | 51.10 | 22.01 | 27.05 | 160.51 | 5.97 |
| $\sigma^2$ | 1.59 | 10.62 | 6.01 | 0.31 | 0.51 | 0.06 |

Table 1: 2-norm statistics on the layers of a population of networks sampled from HyperGAN, compared to 10 standard networks trained from different random initializations. Both HyperGAN and the standard models were trained on MNIST to 99% accuracy. Its easy to see that HyperGAN generates far more diverse networks

the scoring rule $p(y|x) = \frac{1}{N} \sum_{n=1}^{N} p_n(y \mid x, \theta_n)$. It should be noted that we did not perform fine tuning, or any additional training on the sampled networks. The results are shown in Table 2. We generate ensembles of different sizes and compare against both Bayesian (Louizos & Welling, 2016) (Krueger et al., 2017) and non-Bayesian (Lakshminarayanan et al., 2017) methods, as well as MC dropout (Gal & Ghahramani, 2016). We outperform all other methods by using a 100 network ensemble, across all datasets. For all other methods, we report the mean of 100 trials.

| Method | MNIST | MNIST 5000 | CIFAR-5 | CIFAR-10 | CIFAR-10 5000 |
|---|---|---|---|---|---|
| 1 network | 98.64 | 96.69 | 84.50 | 76.32 | 76.31 |
| 5 networks | 98.75 | 97.24 | 85.51 | 76.84 | 76.41 |
| 10 networks | 99.22 | 97.33 | 85.54 | 77.52 | 77.12 |
| 100 networks | **99.31** | **97.71** | **85.81** | **77.71** | **77.38** |
| Deep Ensembles | 99.30 | | 79.00 | | |
| MNFG | 99.30 | | 84.00 | | |
| BHN | 98.63 | 96.51 | | 74.90 | |
| MC Dropout | 98.73 | 95.58 | 84.00 | 72.75 | |

Table 2: Classification performance of HyperGAN on MNIST and CIFAR-10. In order to compare against MNF and Deep Ensembles, we also train a HyperGAN on only the first 5 classes of CIFAR-10, which we denote as CIFAR-5. In addition we examine our generalization ability by training on only 5000 examples of MNIST and CIFAR-10 with a small target network. We do not attempt to outperform state of the art, but we perform better than other probabilistic neural network approaches

### 4.3    1-D Toy Regression Task

We next evaluate the ability of HyperGAN to fit a simple 1D function from noisy samples. This dataset was first proposed by (Hernández-Lobato & Adams, 2015), and consists of a training set of 20 points drawn uniformly from the interval $[-4, 4]$. The targets are given by $y = x^3 + \epsilon$ where $\epsilon \sim \mathcal{N}(0, 3^2)$. We used the same target architecture as in (Hernández-Lobato & Adams, 2015) Lakshminarayanan et al. (2017) and (Louizos & Welling, 2016): a one layer neural network with 100 hidden units and ReLU nonlinearity. For HyperGAN we use two layer generators, and 128 hidden units across all networks. Because this is a small task, we use only a 64 dimensional latent space. MSE loss is used as our target loss function to train HyperGAN.

Results in figure 2 show that HyperGAN clearly learns the target function and captures the variation in the data well. In addition, it can be seen that sampling more networks to compose a larger ensemble improves predictive uncertainty as we sample farther from the mean of the training data.

### 4.4    Anomaly Detection

To test our uncertainty measurements, we perform the same experiments as (Louizos & Welling, 2016) (MNF), (Lakshminarayanan et al., 2017); we measure the total entropy in predictions from HyperGAN-generated networks. We compare our uncertainty measurements with those of MNF and standard ensembles. For MNIST experiments we train a HyperGAN on the MNIST dataset, and test on the notMNIST dataset, which consists of 28x28 binary images of alphabetic letters. In this setting, we want the softmax probabilities on inlier MNIST examples to have maximum entropy - a single large activation close to 1. On off-manifold data we want to have equal probability across predictions. We test our CIFAR-10 model by training on the first 5 classes, and we use the latter

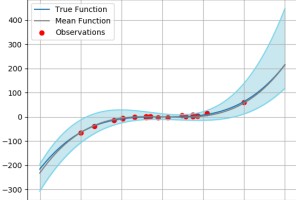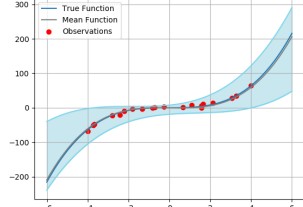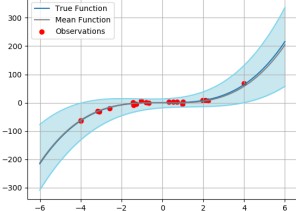

Figure 2: Results of HyperGAN on the 1D regression task. From left to right, we plot the predictive distribution of 10, 100, and 1000 sampled models from a trained HyperGAN. Within each image, the blue line is the target function $x^3$, the red circles show the noisy observations, the grey line is the learned mean function, and the light blue shaded region denotes $\pm 3$ standard deviations

5 classes as out of distribution examples. To build an estimate of the predictive entropy we sample multiple networks from HyperGAN per example, and measure their predictive entropy..

In Fig. 3 we show that HyperGAN can separate CIFAR-10 inlier and outlier samples much better than MNF or standard ensembles. HyperGAN is less certain about data it does not recognize, as the probability of a low entropy prediction is overall lower on outliers. On notMNIST we also show separation, though HyperGAN is also overall less confidant about inliers then MNF. Conventionally trained ensembles without the HyperGAN, referred to as L2 networks in the figure, are highly overconfident on outliers and cannot provide a notion of uncertainty.

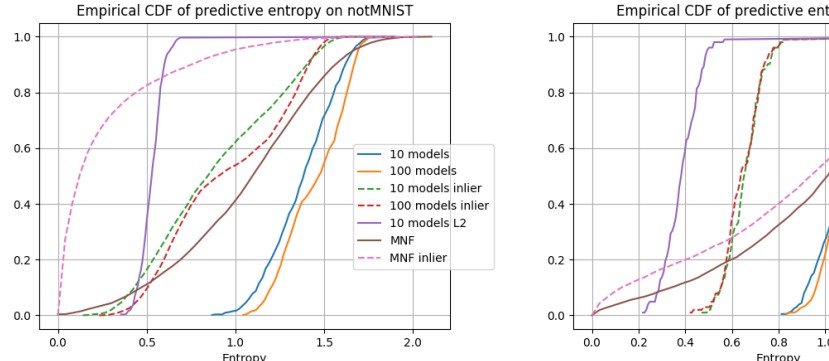

Figure 3: Empirical CDF of the predictive entropy on out of distribution datasets notMNIST, and 5 classes of CIFAR-10 unseen during training. Solid lines denote tests on the respective out of distribution data, while the dashed lines denote entropy on inlier examples (MNIST and CIFAR-10). L2 referres to conventional ensembles trained separately without a HyperGAN

## 4.5 ADVERSARIAL DETECTION

We employ the same experimental setup to the detection of adversarial examples, an extreme sort of off-manifold data. Adversarial examples are often optimized to lie within a small neighborhood of a classifier's decision boundaries. They are created by adding perturbations in the direction of the greatest loss with respect to the model's parameters. Because HyperGAN learns a distribution over parameters, it should be more robust to attacks. We generate adversarial examples using the Fast Gradient Sign method (FGSM) (Goodfellow et al., 2015) and Projected Gradient Descent (PGD) (Madry et al., 2017). FGSM adds a small perturbation $\epsilon$ to the target image in the direction of greatest loss. FGSM is known to underfit to the target model, hence it may transfer well across many similar models. In contrast, PGD takes many steps in the direction of greatest loss, producing a stronger adversarial example, at the risk of overfitting to a single set of parameters. This poses the following challenge: to detect attacks by FGSM and PGD, HyperGAN will need to generate diverse parameters to avoid both attacks. To detect adversarial examples, we first hypothesize that a single adversarial example will not fool the entire space of parameters learned by HyperGAN. If we then evaluate adversarial examples against many generated networks, then we should see a high entropy among predictions (softmax probabilities) for any individual class.

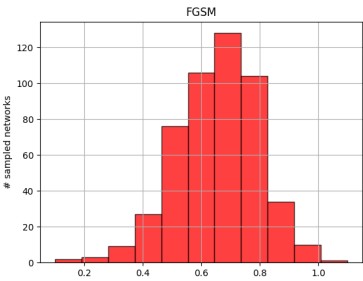 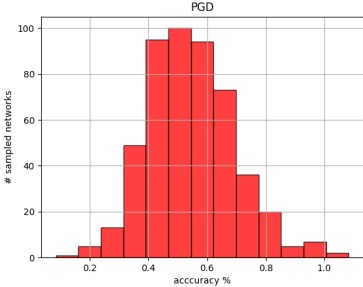

Figure 4: Diversity of predictions on adversarial examples. FGSM and PGD examples are created against a network generated by HyperGAN, and tested on 500 more generated networks. FGSM transfers better than PGD, though both attacks fail to cover the distribution learned by HyperGAN

Adversarial examples have been shown to successfully fool ensembles (Dong et al., 2017), but with HyperGAN one can always generate significantly more models that can be added to the ensemble for the cost of one forward pass, making it hard to attack against. In Fig. 4 we test HyperGAN against adversarial examples generated to fool one network. It is shown that while those examples can fool $50\% - 70\%$ of the networks generated by HyperGAN, they usually do not fool all of them.

We compare the performance of HyperGAN with ensembles of $N \in \{5, 10\}$ models trained on MNIST with normal supervised training. We fuse their logits (unnormalized log probabilities) together as $l(x) = \sum_{n=1}^{N} w_n l_n(x)$ where $w_n$ is the $n$th model weighting, and $l_n$ is the logits of the $n$th model. In all experiments we consider uniformly weighted ensembles. For HyperGAN we simply sample the generators to create as many models as we need, then we fuse their logits together. Specifically we test ensembles with $N \in \{5, 10, 100, 1000\}$ members each. Adversarial examples are generated by attacking the ensemble directly until the generated image completely fools the ensemble. For HyperGAN, we attack the ensemble, but test with a new ensemble of equal size. For MNF we follow their experimental setup and do not form an ensemble. Rather we attack sampled networks and measure the entropy of their predictions.

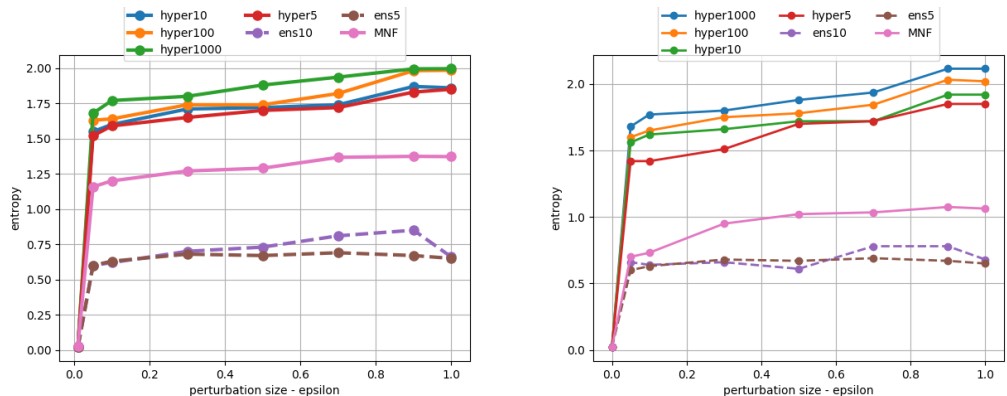

Figure 5: Entropy of predictions on FGSM and PGD adversarial examples. HyperGAN generates ensembles that are far more effective than standard ensembles even with equal population size. Note that for large ensembles, it is hard to find adversarial examples with small norms e.g. $\epsilon = 0.01$

For the purposes of detection, we compute the entropy within the predictive distribution of each of the ensemble members to score the example on the likelihood that it was drawn from the training distribution. Figure 5 shows that HyperGAN clearly identifies adversarial examples as being out-of-distribution, performing better than MNF as well as standard ensembles. HyperGAN is especially suited to this task as adversarial examples are optimized against parameters - parameters which HyperGAN can change. We find that we can successfully detect over 97% of adversarial examples, with a low false positive rate for both attacks just by thresholding the entropy.

## 5 DISCUSSION AND FUTURE DIRECTIONS

We have proposed a generative model for parameter selection which performs strongly on detecting out-of-distribution samples as well as classification. Training a GAN to learn a probability distribution over parameters allows us to non-deterministically sample diverse, performant networks which we can use to form ensembles that can give good uncertainty estimates. Our method is ultimately scalable to any number of networks in the predicting ensemble, requiring just one forward pass to generate a new set of parameters and a low GPU memory footprint. We showed that we can generate models with significant variation over the learned distribution and thus provide uncertainty estimates on outlier data. There is still much room for exploration, we believe that HyperGAN may be useful in a variety of domains including meta learning and reinforcement learning.

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

# A  APPENDIX

## A.1  GENERATED FILTER EXAMPLES

We show the first filter in 25 different networks generated by the HyperGAN to illustrate their difference in Fig. 7. It can be seen that qualitatively HyperGAN learns to generate classifiers with a variety of filters.

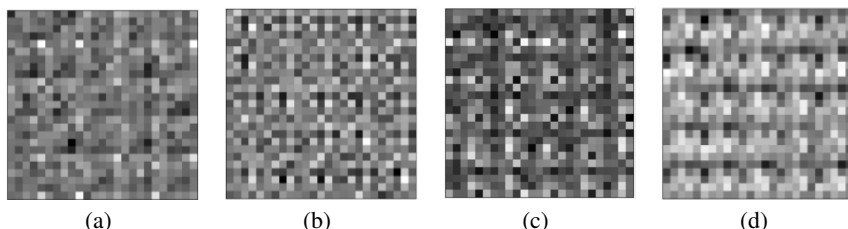

(a)      (b)      (c)      (d)

Figure 6: Convolutional filters from MNIST classifiers sampled from HyperGAN. For each image we sample the same 5x5 filter from 25 separate generated networks. From left to right: figures a and b show the first samples of the first two generated filters for layer 1 respectively. Figures c and d show samples of filters 1 and 2 for layer 2. We can see that qualitatively, HyperGAN learns to generate classifiers with a variety of filters.

## A.2  OUTLIER EXAMPLES

In Figure 7 we show images of examples which do not behave like most of their respective distribution. On top are MNIST images which HyperGAN networks predict to have high entropy. We can see that they are generally ambiguous and do not fit with the rest of the training data. The bottom

row shows notMNIST examples which score with low entropy according to HyperGAN. It can be seen that these examples look like they could come from the MNIST training distribution, making HyperGAN's predictions reasonable

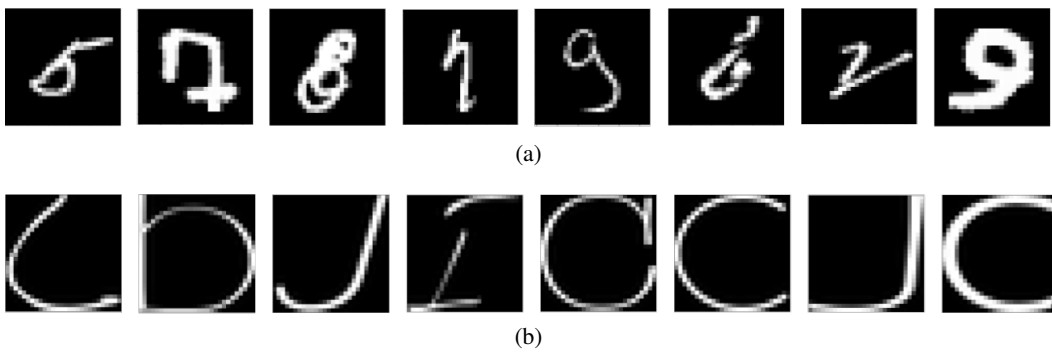

(a)

(b)

Figure 7: Top: MNIST examples to which HyperGAN assigns high entropy (outlier). Bottom: Not-MNIST examples which are predicted with low entropy (inlier)

### A.3 DIVERSITY WITHOUT MIXER OR DISCRIMINATOR

We run experiments on both MNIST and CIFAR-10 where we remove both the mixer and the discriminator. Tables 3 and 4 show statistics of the networks generated by HyperGAN using only independent Gaussian samples to the generators. In this setting, HyperGAN learns to generate only a very small distribution of parameters.

|  | HyperGAN - MNIST | | | Standard Training - MNIST | | |
|---|---|---|---|---|---|---|
|  | Conv1 | Conv2 | Linear | Conv1 | Conv2 | Linear |
| Mean | 10.79 | 106.39 | 14.81 | 27.05 | 160.51 | 5.97 |
| $\sigma^2$ | 0.58 | 0.90 | 0.79 | 0.31 | 0.51 | 0.06 |

Table 3: Statistics on the layers of a population of networks sampled from HyperGAN, compared to 10 standard networks trained from different random initializations. Without the mixing network or the discriminator, HyperGAN suffers from a lack of diversity

| HyperGAN - CIFAR-10 | | | | | |
|---|---|---|---|---|---|
|  | Conv1 | Conv2 | Conv3 | Linear1 | Linear2 |
| Mean | 1.87 | 16.83 | 9.35 | 10.66 | 20.35 |
| $\sigma^2$ | 0.11 | 2.44 | 1.02 | 0.16 | 0.76 |
| Standard Training - CIFAR-10 | | | | | |
|  | Conv1 | Conv2 | Conv3 | Linear1 | Linear2 |
| Mean | 5.13 | 15.19 | 16.15 | 11.79 | 2.45 |
| $\sigma^2$ | 1.19 | 4.40 | 4.28 | 2.80 | 0.13 |

Table 4: Statistics on the layers of networks sampled from HyperGAN without the mixing network or discriminator, compared to 10 standard networks trained from different random initializations

### A.4 HYPERGAN NETWORK DETAILS

In tables 5 and 6 we show how the latent points are transformed through the generators to become a full layer of parameters. For a MNIST based HyperGAN we generate layers from small latent points

of dimensionality 128. For CIFAR-10 based HyperGANs we use a larger dimensionality of 256 for the latent points.

<table>
<tr><td colspan="3">Table 5: MNIST HyperGAN Target Size</td></tr>
<tr><th>Layer</th><th>Latent size</th><th>Output Layer Size</th></tr>
<tr><td>Conv 1</td><td>128 x 1</td><td>32 x 1 x 5 x 5</td></tr>
<tr><td>Conv 2</td><td>128 x 1</td><td>32 x 32 x 5 x 5</td></tr>
<tr><td>Linear</td><td>128 x 1</td><td>512 x 10</td></tr>
</table>

| Layer | Latent size | Output Layer Size |
|-------|-------------|-------------------|
| Conv 1 | 128 x 1 | 32 x 1 x 5 x 5 |
| Conv 2 | 128 x 1 | 32 x 32 x 5 x 5 |
| Linear | 128 x 1 | 512 x 10 |

Table 5: MNIST HyperGAN Target Size

| Layer | Latent Size | Output Layer Size |
|-------|-------------|-------------------|
| Conv 1 | 256 x 1 | 16 x 3 x 3 x 3 |
| Conv 2 | 256 x 1 | 32 x 16 x 3 x 3 |
| Conv 3 | 256 x 1 | 32 x 64 x 3 x 3 |
| Linear 1 | 256 x 1 | 256 x 128 |
| Linear 2 | 256 x 1 | 128 x 10 |

Table 6: CIFAR-10 HyperGAN Target Size

## A.5 CONNECTIONS TO GANS AND WASSERSTEIN AUTO-ENCODERS

Despite being largely applied to the same task of learning probability distributions, implicit generative models such as Generative Adversarial Networks (GAN) and Wasserstein Auto-encoders (WAE) are presented under different frameworks. GANs are a likelihood free density estimator, specifying only a generator, and do not have any notion of reconstruction. WAEs find an optimal transport map between target and model distributions by first encoding samples to a latent space and reconstructing the original data samples using a generator. Even with these differences, many combinations of these models have been proposed. Creating composite models and loss functions from the GAN and WAE building blocks is commonplace.

HyperGAN is also an implicit generative model, in that there is no explicit data distribution from which to sample from. HyperGAN learns to estimate from the data distribution not from real samples, but from the error of its estimates. We believe that HyperGAN is not just a combination of existing generative models. To show this we give an overview of both GANs and WAE, and contrast HyperGAN to them.

### GENERATIVE ADVERSARIAL NETWORKS

GANs in the original formulation (Goodfellow et al., 2014) play a minimax game between two neural networks. A generator $G : Z \rightarrow X$ samples a prior distribution $P_Z$ on the latent space $Z$ and attempts to produce outputs from the data distribution $X$. A discriminator $D$ evaluates samples from $X$ and from $G(Z)$ and outputs a score between $[0, 1]$. This score given by the discriminator corresponds to how likely the input is to have come from $X$, according to $D$. This creates the following minimax game between the generator and the discriminator:

$$\min_G \max_D \mathbb{E}_{x \sim X}[\log D(x)] + \mathbb{E}_{z \sim P_Z}[\log 1 - D(G(z))] \tag{7}$$

The goal of the generator is to learn to transform noise samples into samples which fool the discriminator, thus minimizing the loss on $G$. While the discriminator tries to maximize its ability to tell real images from fake samples from the generator.

In contrast, HyperGAN is not a likelihood free model like a GAN. While GANs do not specify an explicit evaluation criterion, HyperGAN minimizes the negative likelihood of the generated parameters with an explicit criterion: the loss on the target task. HyperGAN then has the benefit of a principled quantitative evaluation of its samples, and does not need to rely on metrics like the Inception (Salimans et al., 2016) or FID scores (Heusel et al., 2017) commonly used with GANs.

### WASSERSTEIN AUTO-ENCODERS

Wasserstein Auto-encoders (WAE) (Tolstikhin et al., 2017) use a similar formulation under the optimal transport framework (Villani, 2008) to minimize the difference between generative model $P_G$ and the data distribution $P_X$, where $X \in \mathcal{X}$ in this case are images. In the WAE, A latent space $Z$ and prior distribution $P_Z$ on $Z$ are defined as sample spaces. In general, there is an optimal transport map between $P_X$ and $P_G$ if the data distribution $P_X$ is well distributed according to a prior distribution (in this case $P_Z$). The problem is that the data distribution $P_X$ is not a well defined distribution, and so computation of an OT mapping is intractable. To remedy this, an encoder $Q : \mathcal{X} \rightarrow Z$ is used to project points from the data distribution to the latent space. WAE considers the marginal

distribution $Z \sim Q(Z|X)$. The conditional $Q(Z|X)$ represents the *aggregated posterior $Q_Z$*: the space of $Z$ which we can optimize over with an encoder. The authors argue that this allows WAE to not have to search over all possible maps $P_X \rightarrow P_G$, but only those originating from the better behaved $Q_Z$. The encoder's aggregated posterior $Q_Z$ still needs to be distributed according to prior $P_Z$, so a penalty is imposed on $Q_Z$ to match the mean and covariance of $P_Z$. In principle the penalty term $\mathcal{D}_Z(P_Z, Q_Z)$ could be any distance between probability distributions. WAE uses either MMD or a discriminator trained with an adversarial loss.

$$\inf_G \inf_{Q(Z|X) \in \mathcal{Q}} \mathbb{E}_{P_X} \mathbb{E}_{Q(Z|X)} \left[ c(X, G(Z)) \right] + \lambda \mathcal{D}_Z(P_Z, Q_Z) \tag{8}$$

Where $c$ is any measurable cost function measuring the error between the data $X$ and reconstruction $G(Z)$, and $\lambda$ is a hyperparameter.

While HyperGAN and WAE use a similar architecture, HyperGAN is not an auto-encoder. As such, HyperGAN replaces the WAE reconstruction cost function $c(X, G(Z))$ with a term that does not depend on $X$. Instead, $X$ is hidden, and HyperGAN minimizes the KL divergence between estimates $\theta_G \in \Theta_G$ and the true parameter distribution $\Theta$. Because the Wasserstein distance does not reduce to the KL divergence, we can not exactly compute the optimal transport map:

$$\inf_G \inf_{Q(Z|S) \in \mathcal{Q}} c[S, G(Z)] \tag{9}$$

because we cannot compute the reconstruction cost $c$. To compute $c$ we need samples from the target distribution. Instead, we can only approximate the optimal map by minimizing

$$\inf_G \inf_{Q(Z|S) \in \mathcal{Q}} \mathcal{L}(\mathcal{F}(X; G(Z), Y)). \tag{10}$$

Which nonetheless gives us well-behaved samples from a learned distribution of parameters.

