# OpenReview forum: "HyperGAN:  Exploring the Manifold of Neural Networks"
_ICLR.cc/2019/Conference_

### Official Review · AnonReviewer1 · 2018-10-29
**Nice idea**

**Rating:** 4
**Confidence:** 3

**Review:**

TL;DR. I find the manuscript to contain interesting ideas, yet I believe there is room for improvement.

* Summary

For any given specific network architecture, the manuscript aims at learning a distribution over the weights (rather than point-wise estimates of the weights). This is achieved through using a two-steps procedure, in which a "hypernetwork" is trained to output weights for the network of interest, and a GAN is then used to (adversarially) generate samples from a distribution $Q$ which is assumed not too far (in a KL sense) from a Gaussian prior $P$.

* Major issues

I find the central idea to be of interest to the ICLR community. However I have found a number of shortcomings to be addressed before I could recommend acceptance. The following list is in no particular order.

- References: 20 out of 22 (!) references are preprints, about half of which are 3+ years old. Most of them are now published in proceedings and I strongly feel proper credit should be given to authors by replacing outdated preprints with correct citations.
- Links with Bayesian deep learning: I feel this should be more carefully discussed in the manuscript. The sentence "We have proposed a generative, non-Bayesian solution [...]" should be explained, as from what I gather HyperGAN samples (GAN-like) weights (i.e., networks) from a distribution $Q$ which is deemed not too far (in the KL sense) from a prior distribution $P$. How is that not Bayesian?
- Numerical experiments. Table 4: what are the numbers reported? If single evaluation, I do not believe any conclusion may be drawn. If averages over multiple repetitions, no conclusion can be drawn without reporting standard deviations. In addition, I do not quite grasp the purpose of the 1D toy example.
- Overall, I think the authors should try and make their contributions and method clearer. For example, a pseudo-code of the whole procedure might help readers understand the gist.
- Architecture specific: I find the claim that HyperGAN explores the manifold of neural nets too strong. As the whole procedure is architecture-specific, I would find more appropriate to change that claim to "exploring the weights distribution for a specific architecture".
- Code availability: the scope of the paper is diminished by the fact that no code is available by the time of review. A toolbox (not disclosing the authors' identities) should be made available to support the manuscript claims. Last sentence (page 8) is likely to be outdated and should be removed.

* Minor issues

- some typos: architecture (caption figure 1), $G(Q(z))$ (missing parenthesis, page 3), sum index $n$ not used in the last equation (page 7).
- "Perhaps the first proposed method..." (page 2). Such imprecise statements must be avoided.

---

> ### Author Response · Authors · 2018-11-26
> **Reply**
>
> We have updated the references and replaced them with conference proceedings where applicable. We have updated section 3 to make the technical description more clear.
>
> Question: Connection and difference with Variational inference
> HyperGAN looks on the surface similar to Bayesian methods like variational inference, in implicit generative models like VAE, VI is done by minimizing the KL divergence between the prior and the latent space. This is done by assuming a Gaussian encoder and prior so that the KL term can be computed. Instead of using a closed-form distribution as in VI, we enforce the distributional constraint on Q by using a discriminator trained adversarially. We have removed the language stating that our method is not Bayesian. It’s more accurate that we don’t make use of fully probabilistic models as in our model there is no encoder that goes from network parameters to the latent sample, and as in other VAE/GAN-based approaches, we can’t compute the probability of generating a particular set of parameters.  Our rewritten Section 3 makes this more clear.
>
> Question: What’s expected out of the 1D regression task?
> The 1D toy regression task is a simple test which shows that HyperGAN can learn more than just the mean function of the training data. In any model which can measure uncertainty, we would expect the model to give a wider distribution of predictions as we move farther from the training data. We can see that this happens reliably with HyperGAN-generated ensembles of different sizes.
>
> Question: Title is inaccurate
> We have edited the title to reflect that we are generating diverse neural networks, instead of a whole manifold.
>
> Question: Code is not available
> The code is now available on Github at : https://github.com/ICLR19HyperGAN/HyperGAN

---

### Official Review · AnonReviewer3 · 2018-10-30
**Good experimental results but lacking rigour**

**Rating:** 5
**Confidence:** 4

**Review:**

This works propose a new approach to learn to sample (or generate) the parameters of a deep neural networks to solve a task. They propose a new architecture inspired by hyper networks and adversarial auto-encoders, where the parameters of the networks are generated from a low dimensional latent space. By using an ensemble of networks sampled with their approach they're able to get state of the art results on uncertainty estimation.

The notations are confusing and the paper contains several mistakes. In particular:
- P_z is used to represent different distributions. It sometimes refers to the distribution of the latent variables and sometimes to the prior over the weight embeddings. Different notation should be used to represent different quantity.
- D_z sometimes refers to the regularization term or to the discriminator.
- Eq 2. I believe there is a bug in the equation, the expectation is over Q(z) but it should be P_z (distribution of the latent variable z), otherwise it doesn't make much sense.
- The equation for the cross entropy is wrong. If y_i are the true labels and F(x_i, theta) is the prediction then it should be y_i*log(F(x_i, theta)).
- It's not clear if the loss of the discriminator should be maximized for the parameters of the discriminator and minimized with respect to the parameters of the encoder. Furthermore it would be interesting to study what is the impact of this particular choice of loss for the discriminator. In particular I invite the author to compare the loss proposed to the loss in [1].
Fixing these, would make the paper much easier to understand.

The authors motivates their approach by drawing a link with wasserstein (WAE) and adversarial auto-encoders. While this could be interesting I think this link should be made more formal.
Indeed, the WAE is derived from the wasserstein distance between the true data distribution and the distribution of the model. However it's not clear if the approach proposed can still be derived from such a principle. I would invite the author to make the link between wasserstein distance minimization and their approach more explicit.

To my knowledge the method proposed is novel, however using implicit posterior to learn the weights is not novel and several other works have looked at it. In particular I think [1,2] should be discussed in the related work.
The difference with traditional bayesian approach such as variational inference should also be discussed, since the approach is really close to approximating the posterior with an implicit distribution and computing the KL term using a GAN (like in [3,4]).

I think one interesting novelty that needs to be emphasized is that the model has both: parameters that are point estimates (the parameters of the generators) and parameters that are sampled from a posterior distribution (the weight embeddings).

Pros:
- Good and promising experimental results.

Cons:
- The paper combines several tricks and ideas but it's not really clear what is important and why such an approach works. For example how important is the latent space and the encoder ? Could we just sample directly the weight embeddings from a gaussian and remove the regularization ?
- The other points mentioned above about the clarity of the paper.

Others:
- The title is misleading, the manifold is not really explored... If the author really want to explore the manifold some interesting questions are:  what happens if we try to interpolate between two latent variables ? What do the latent variables represent ? what's the influence of the dimension of the latent space ?
- In the experiments: what is the number of networks used for the other methods ?
- It would be nice to have a plot showing the accuracy as a function of the perturbation in section 4.5.

Conclusion:
The experimental results seem promising however the motivation for the approach is not clear. I think fixing some of the points mentioned above could greatly improve the clarity of the paper and make it a stronger submission. In the current state I don't believe the paper is rigorous enough to be accepted.

References:
[1] Pawlowski, N., Rajchl, M., & Glocker, B. (2017). Implicit weight uncertainty in neural networks. arXiv:1711.01297.
[2] Wang, K. C., Vicol, P., Lucas, J., Gu, L., Grosse, R., & Zemel, R. (2018, July). Adversarial Distillation of Bayesian Neural Network Posteriors. ICML
[3] Mescheder, L., Nowozin, S., & Geiger, A. (2017, July). Adversarial Variational Bayes: Unifying Variational Autoencoders and Generative Adversarial Networks. ICML
[4] Huszár, F. (2017). Variational inference using implicit distributions. arXiv:1702.08235.

---

> ### Author Response · Authors · 2018-11-26
> **Reply**
>
> We have revised the manuscript and addressed the notation and terminology concerns as well as the comments made by other reviewers. We have improved these to better describe our method. We would really appreciate if you can read the new Section 3 and give us a new evaluation and further feedback. With the revision, we still want to answer the questions laid out here.
>
> Question: The difference between a traditional Bayesian approach such as variational inference should also be discussed. It would be interesting to study what is the impact of this particular choice of loss for the discriminator. In particular, I invite the author to compare the loss proposed to the loss in [1].
> Answer:  From the new description one can see that there is a significant difference between our approach and variational Bayesian approach in that we never explicitly model the KL-divergence term that comes from p(z|\theta), because we do not have explicit theta samples nor had an encoder.
> In Bayes by Hypernetwork (BbH) [1], we note two differences. First, the prior matching step treats each weight independently. This is different from HyperGAN where we perform the prior matching step between the prior and the continuous mixture Q(s) using the adversarial loss. Second, BbH uses independent noise samples as input to the generators. We found that this configuration hurt the diversity of our generated networks, which is why we use the mixer Q to introduce correlations to our single noise sample. In Table 3 and 4 we compare against using independent generators and find that we lose significant diversity in our generated ensembles.
>
> Question: What is the number of networks used for other methods?
> Answer: Each other (non-HyperGAN) method in Table 2 uses the mean of 100 samples, which corresponds to our strongest considered ensemble of 100 networks. We only used 10 networks for the L2 (standard) ensembles in the adversarial detection experiments in Sec. 4.5, because of the prohibitive cost of training 100 neural networks on each task.
>
> Question: It would be nice to have a plot showing the accuracy as a function of the perturbation in section 4.5.
> Answer: We tested only on adversarial examples which succeeded in fooling our ensemble. This means the accuracy of the ensemble predictions under all perturbation levels is 0, so we chose not to plot it.
>
> Question: Title is inaccurate
> We have also edited the title to reflect that we are generating diverse neural networks, instead of a whole manifold.
>
> [1] Pawlowski, Nick, et al. "Implicit weight uncertainty in neural networks." arXiv preprint arXiv:1711.01297 (2017).

---

> > ### Comment · AnonReviewer3 · 2018-12-03
> > **Reply**
> >
> > Thanks for the clarifications, the section 3 has been greatly improved (thus I slightly increased my score) but there is still room for improvement.
> >
> > 1. I think the section 3.1 is unclear and potentially unnecessary. It is well known that minimizing the KL between the true data distribution and the model is equivalent to maximizing the log likelihood of the model. Plus there is a mistake in equation 6 the expectation should be taken over p(x|\theta).
> > 2. I'm not sure to understand table 3 & 4, do you sample the q_n directly from P ? If so I don't understand why this would "collapse" ? since you actually argue in section 3 that "This constraint makes it closer to the generated parameters and ensures that Q(s) itself does not collapse to always outputting the same latent code".
> >
> > As a note I recommend for next time that you don't share your code through a GitHub link as this can compromise anonymity.

---

### Official Review · AnonReviewer2 · 2018-10-30

**Rating:** 6
**Confidence:** 5

**Review:**

This paper proposes a technique for learning a distribution over parameters of a neural network such that samples from the distribution correspond to performant networks. The approach effectively encourages sampled parameters to have low loss on the training set, and also uses an adversarial loss to encourage the distribution of parameters to be Gaussian distributed. This approach can improve performance slightly by using ensembling and can be useful for uncertainty estimates for out-of-distribution examples. The approach is tested on a few simple problems and is shown to work well.

I am definitely in favor of exploring adversarial divergences (using a critic as a differentiable loss to compare two distributions) in unusual settings, and this paper certainly does this. The idea of transforming samples from a prior such that the transformed sample corresponds to useful network parameters is interesting. The results also seem promising. However, currently the mathematical description of this method is completely unclear and ridden with many errors. I can understand at a reasonable level what the approach is doing from Figure 1, but the definitions and equations given in Equation 3 are at times nearly incomprehensible "mathiness". I'm giving the paper a borderline accept because the idea is interesting and the results are OK; I will raise my score if Section 3 is dramatically improved. I give some specific examples of issues with Section 3 in my specific comments below. I'd also note that the paper does a somewhat poor job comparing to existing work - only section 4.2 includes a comparison to existing "uncertainty" methods. This should also be improved - the authors should implement the existing methods and use them as a point of comparison in all of their experiments. As a final high-level note, the approach is described at various points as an "autoencoder" particularly in reference to the adversarial autoencoder. However, the approach does not "autoencode" anything - there is no reconstruction term, or input apart from the noise samples. The only thing it has in common with the adversarial autoencoder is the use of a critic to enforce a distributional constraint. Calling it, or comparing it to, an autoencoder is confusing and misleading.

Specific comments:

- You mention fast weights in related work. I believe Hinton and Plaut were the first to propose fast weights in "Using Fast Weights to Deblur Old Memories", and I'd also suggest mentioning "Using Fast Weights to Attend to the Recent Past" which is a more recent demonstration that fast weights can be useful on modern problems.
- The are some issues with your description of Equation 1: First, I don't believe you define G(z) (I assume it is the "decoder" network; please define it). Second, in practice I don't believe you actually use JSD or MMD for D_z; you use a critic architecture which in some limit approximates some statistical divergence but in practice they typically don't (see e.g. Arjovsky and Bottou 2017; Fedus et al. 2017; Rosca et al. 2018). Third, writing Q_z \sim Q(z | x) seems strange to me - Q_z is a distribution, and I don't believe that Q(z | x) is a distribution over distributions, so how are you sampling a distribution (Q_z) from Q(z | x) as suggested by the use of the \sim notation? I think you simply mean that Q_z is Q(z | x) approximately marginalized over x.
- Equation 2 is also not clear. First, the sentence before starts "Suppose the real parameters \theta^* \sim \Theta..." The equation itself does not include \theta^* or \Theta so I don't see what this is referring to. Second, the expression for an m-dimensional is written \mathcal{N}(0, \sigma^2, I_m). It's not clear why there is a comma before I_m, and I_m is not defined (though I assume it is the m \times m identity matrix) - did you mean to multiply I_m by \sigma^2? Third, it looks like you actually define P_z twice, once as "an m-dimensional isotropic Gaussian" and again as "a Kd-dimensional isotropic Gaussian"; am I to infer that m = Kd? Why use both? Fourth, you mention the joint P(x, y) but the expectation is taken over P_x and P(y | x). Why call it P_x and not P(x)? And why not compute the expectation over P(x, y)? Fifth, you write "Here the encoder..." -- you never define that Q(z) or G is "the encoder", I assume Q(z). It is strange to take the expectation over Q(z) (I assume sampling z \sim Q(z)) but then have the term Q(z) appear in (2). How are Q(z) and Q_z related? On that note, I don't see how (2) is an autoencoder, since there is no Q(z | x) term. It appears instead that you are sampling z from Q(z) which doesn't condition on x. So what is being autoencoded? Related, you write "all the q_k (that will generate different layers) will be correlated, unlike dimensions of z which are drawn to be independent from each other." But if Q(z) = [q_1, ..., q_K] then doesn't the secont term in (2) suggest that they are being enforced to be similar to the prior P_z, and therefore uncorrelated? Note that you also say later on "The job of the regularizer D_z(P_z, Q_z) is to force each embedding q_n to approximate P_z." Frankly at this point I will stop pointing out issues with this equation and discussion since they are so widespread.
- In your definition of your ensemble scoring rule, you are taking the sum over N + 1 elements (n = 0 to N) but dividing by N.
- In 4.2, do you use the same model architecture/training/regularization etc. as in previous studies? If not I think comparing the different methods will be conflated by differences in training procedures. Since you do not report results in many experimental settings, I assume you don't.
- In Figure 3, why not plot the true standard deviation around the true function? It appears you are only plotting +/- 3 stndard deviations for the learned function.
- Why not include 100 models L2 on Figure 3?
- It's not clear to me why you define your "disagreement d" when it appears the same as the entropy score you used in 4.4.
- A stronger and more convincing attack would be to attack the ensemble of models, instead of attacking a single model and testing on the ensemble.

---

> ### Author Response · Authors · 2018-11-26
> **Reply**
>
> Thanks a lot for your valuable comments.
> We have thoroughly reworked the manuscript, especially section 3, to address these concerns as well as those had by other reviewers. We would really appreciate if you can read the new Section 3 and give us a new evaluation and further feedback.
> We want to address some specific issues here:
>
> -We have added comparisons to [1], as well as discussion of the results. We conducted these experiments using model architectures identical to our own. We will add [2] and [3] in the next version of the paper,
>
> -We have added the suggested citations to our related work section
>
> -We have reworked our treatment of “auto-encoders”. We have re-written Section 3 to point out the similarity and differences between HyperGAN and conventional GANs. Indeed, HyperGAN is not an auto-encoder since nothing was encoded explicitly. Hence we changed all the “encoder” text in the manuscript to a new term called mixer. The mixer serves to introduce dependencies in the latent samples, which turned out to be crucial for the diversity of the networks generated by HyperGAN (Table 3, 4). We have added a section in the appendix which thoroughly explains these methods and how HyperGAN differs from both.
>
> -We did not use ensembles of 100 (non-HyperGAN) models for our anomaly detection because it becomes computationally prohibitive for us to train so many models, especially for CIFAR-10. One of the main advantages of HyperGAN is to generate hundreds and thousands of models effortlessly from the generator without additional training, while it is time-consuming to generate conventional L2 ensembles with more than a handful of models. We note that there is a drastic difference between HyperGAN and L2 ensembles with 10 models (Fig. 3 and Fig. 5).
>
> -In the adversarial attack experiment (section 4.5), we indeed attacked our ensembles (both standard and HyperGAN) until we have adversarial examples which fooled the ensemble for Fig. 5. There was a typo in the previous paragraph where we mentioned fooling just one network and test it on the whole ensemble, which should refer to Fig. 4 and is a separate experiment that we did to show the validity of using the uncertainty to detect adversarial examples. We have revised the explanation in the manuscript to make this more clear.
>
> [1] Louizos, Christos, and Max Welling. "Multiplicative normalizing flows for variational bayesian neural networks." arXiv preprint arXiv:1703.01961 (2017).
> [2] Wang, K. C., Vicol, P., Lucas, J., Gu, L., Grosse, R., & Zemel, R. (2018, July). Adversarial Distillation of Bayesian Neural Network Posteriors. ICML
> [3] Gal, Yarin, and Zoubin Ghahramani. "Dropout as a Bayesian approximation: Representing model uncertainty in deep learning." international conference on machine learning. 2016.

---

> > ### Comment · AnonReviewer2 · 2018-11-29
> > **Re: Reply**
> >
> > Thank you for addressing my comments and for your updates. Section 3 is clearly improved; I believe it is more clear what the model is now although since I spent so much time trying to figure out what was going on in the first draft I may have an easier time understanding it than someone who is being exposed to the idea for the first time. The paper remains somewhat weak in terms of comparing to existing work, though I appreciate the inclusion of [1] as a baseline. I have raised my score accordingly but I would suggest the authors continue to rework the paper, improve the baselines, and resubmit it to another conference if they are interested in it being published.
> >
> > A remaining low-level question (no need to answer explicitly here): Why are so many entries in Table 2 empty? Did the authors reimplement the baselines they are comparing against? If you are just quoting numbers from previous papers, I bet that your implementation of the model and their implementation is quite different and the results cannot be reliably compared. I would think you would need to reimplement all of these baselines in the same framework/model implementation/training scheme/etc. to get a reliable comparison.

---

### Public Comment · ~James_Lucas1 · 2018-10-04
**A closely related paper**

Previous work has addressed parameter generation using GANs. Please check out this paper: http://proceedings.mlr.press/v80/wang18i.html

In my opinion, this work still has novelty but a discussion/comparison seems due.

---

> ### Author Response · Authors · 2018-10-07
> **Thanks for the input about related work**
>
> Thank you for alerting us of this recent work we missed. We will be sure to cite and compare against it in the final paper.
> Our paper is, although, quite different from the aforementioned reference [1]. The offline setting in [1] is the approach we considered and didn't take (page 3) - to first train many networks and train a GAN on them. In the online setting in [1], a GAN is trained from samples taken from a single training procedure, with noise added from MCMC. The MCMC samples and GAN updates are interleaved. With this approach, the parameters that are used to train the GAN are highly correlated with each other and the diversity of the networks that the trained GANs can generate is questionable.
>
> In contrast, HyperGAN generates networks purely from random noise samples. Our approach doesn't use correlated examples in training and hence we believe it can generate more diverse networks (Table 3). We also show that HyperGAN can achieve higher classification accuracy on testing data and ensembles of models help significantly, showcasing the diversity of generated networks.
>
> [1] Wang, K.-C., Vicol, P., Lucas, J., Gu, L., Grosse, R., and Zemel, R. Adversarial Distillation of Bayesian Neural
> Network Posteriors. In Proceedings of the 35th International Conference on Machine Learning, 2018

---

### Meta-Review · Area_Chair1 · 2018-12-14
**Interesting work that requires a bit of fine tuning.**

**Confidence:** 4
**Recommendation:** Reject

**Metareview:**

All of the reviewers find this paper to contain interesting ideas. Originally, clarity was a major issue, although a few issues remain (see the comments of reviewer 3). The reviewers believe that the paper has been substantially improved from its original form, however there is still room for improvement: more comprehensive comparisons to existing work (reviewer 1), careful ablations (reviewer 3), etc. With a little bit of polish, this paper is likely to be accepted at another venue.

I am certainly not penalizing you for anonymously sharing your code on Github, as this was specifically requested by reviewer 1.